# Multiplexed Host-Induced Gene Silencing of *Aspergillus flavus* Genes Confers Aflatoxin Resistance in Groundnut

**DOI:** 10.3390/toxins15050319

**Published:** 2023-05-05

**Authors:** Kalyani Prasad, Kalenahalli Yogendra, Hemalatha Sanivarapu, Kanniah Rajasekaran, Jeffrey W. Cary, Kiran K. Sharma, Pooja Bhatnagar-Mathur

**Affiliations:** 1International Crops Research Institute for the Semi-Arid Tropics (ICRISAT), Hyderabad 502324, India; kalyanip36@gmail.com (K.P.); yogendra.kalenahalli@icrisat.org (K.Y.);; 2Southern Regional Research Center, Agricultural Research Service, United States Department of Agriculture (USDA/ARS), New Orleans, LA 70124, USA; kanniah.rajasekaran@usda.gov (K.R.); jeff.cary@usda.gov (J.W.C.); 3Sustainable Agriculture Program, The Energy and Resources Institute (TERI), India Habitat Center, New Delhi 110003, India; 4International Maize and Wheat Improvement Center (CIMMYT), El Batán, Texcoco 56237, Mexico

**Keywords:** aflatoxin, *Arachis hypogaea* L., *Aspergillus flavus*, fatty acid, host-induced gene silencing, groundnut, proteomics

## Abstract

Aflatoxins are immunosuppressive and carcinogenic secondary metabolites, produced by the filamentous ascomycete *Aspergillus flavus*, that are hazardous to animal and human health. In this study, we show that multiplexed host-induced gene silencing (HIGS) of *Aspergillus flavus* genes essential for fungal sporulation and aflatoxin production (*nsdC*, *veA*, *aflR*, and *aflM)* confers enhanced resistance to *Aspergillus* infection and aflatoxin contamination in groundnut (<20 ppb). Comparative proteomic analysis of contrasting groundnut genotypes (WT and near-isogenic HIGS lines) supported a better understanding of the molecular processes underlying the induced resistance and identified several groundnut metabolites that might play a significant role in resistance to *Aspergillus* infection and aflatoxin contamination. Fungal differentiation and pathogenicity proteins, including calmodulin, transcriptional activator-HacA, kynurenine 3-monooxygenase 2, VeA, VelC, and several aflatoxin pathway biosynthetic enzymes, were downregulated in *Aspergillus* infecting the HIGS lines. Additionally, in the resistant HIGS lines, a number of host resistance proteins associated with fatty acid metabolism were strongly induced, including phosphatidylinositol phosphate kinase, lysophosphatidic acyltransferase-5, palmitoyl-monogalactosyldiacylglycerol Δ-7 desaturase, ceramide kinase-related protein, sphingolipid Δ-8 desaturase, and phospholipase-D. Combined, this knowledge can be used for groundnut pre-breeding and breeding programs to provide a safe and secure food supply.

## 1. Introduction

*Aspergillus flavus* is a ubiquitous saprophytic fungus that infects maize, groundnut, cotton, chilies, and several nuts and seed crops [1,2]. Infection by *A. flavus* leads to the production of carcinogenic secondary metabolites, including polyketide-derived aflatoxins [3], that are serious health hazards to humans and animals, leading to an annual loss of over USD 932 million globally [3,4]. Groundnut (*Arachis hypogaea* L.) is highly vulnerable to *Aspergillus* invasion and aflatoxin contamination, which not only poses a health hazard, but also hampers international trade [5,6]. Despite numerous breeding efforts, lack of the resistance germplasm has been a limiting factor in achieving significant progress so far [7].

The emergence of biotechnological methods offers a novel and environmentally safe approach to obtaining aflatoxin-resistant groundnuts [8,9]. Based on the understanding of molecular patterns underlying plant–*A. flavus* interactions, host-induced gene silencing (HIGS) has proven to be an effective approach due to enhanced trait durability, as the host plant acts as a delivery system to induce gene silencing in *A. flavus* [9,10]. Recent studies have discovered the genes involved in each step of the aflatoxin biosynthetic pathway [11,12]. Targeted downregulation of the aflatoxin biosynthetic genes *aflR* [13], *aflM* [14], and *aflC* [10] in maize and *aflM* and *aflP* [9] and five RNAi genes, *aflR*, *aflS*, *aflC*, *pes1*, and *aflep* [8,15], in groundnut can provide considerable success in developing aflatoxin-resistant genotypes. Moreover, aflatoxin biosynthetic pathway genes and silencing of *Aspergillus* genes, including alkaline protease (*alk*) [16] and alpha-amylase (*amy1*) [17], have been reported in maize.

Here, we describe an improved host-induced gene silencing strategy in groundnut to simultaneously control fungal infection and aflatoxin contamination by multiplexed silencing of four *A. flavus* genes, such as *nsdC* [18] and *veA* [19], involved in fungal developmental processes, including conidiophore biogenesis, sclerotial production, and aflatoxin production—*aflR* [20] transcriptional regulation of aflatoxin production and *aflM* (*Ver1*) [21], an aflatoxin biosynthetic pathway clustered gene that converts Versicolorin A (VERA) to Demethylsterigmatocystin (DMST). We were able to obtain groundnut transformants that had significantly lower *A. flavus* infection, while showing an enhanced reduction in aflatoxin contamination within safer levels. Furthermore, to develop effective control strategies for aflatoxin resistance, we used liquid chromatography coupled with hybrid mass spectrometry (LC-MS/MS)-based non-target proteomics of two contrasting groundnut lines (resistant HIGS and susceptible wild type (WT)) infected by *A. flavus* to better understand the underlying resistance mechanism to fungal infection and aflatoxin contamination at the molecular level. The findings provide significant insights that enable comparison of both genotypic and time points of post-harvest groundnut–*A. flavus* interaction and unravel the mechanisms that provide resistance and possibly susceptibility in HIGS lines expressing the four RNAi genes.

## 2. Results

### 2.1. Generation of HIGS Lines Overexpressing 4RNAi Cassette

To knock down *A. flavus* developmental and aflatoxin biosynthetic cluster genes that could be processed by the host’s RNAi machinery (Figure 1A,B), HIGS lines overexpressing the 4RNAi genes’ inverted repeat sequences were developed by *Agrobacterium*-mediated transformation of cv. ICGV 91114. Overall, 44 putative primary transformants (T_0_) were successfully produced, and the presence of the transgene was confirmed by PCR analysis using 4RNAi primers (Figure 1C). Eleven randomly selected groundnut HIGS events in the T_3_ generation revealed a single-copy integration of the *aflR* gene in all tested events (Appendix A). Further, inheritance analysis indicated the integration of all 4RNAi genes in the groundnut genome in a 3:1 segregation ratio (Appendix A).

Reverse transcription PCR analysis of homozygous T_3_ lines revealed active transcription of the 4RNAi gene cassette in the HIGS events, while no transcripts were detected in the WT control lines (Figure 1D). Quantitative RT-PCR (qRT-PCR) assays using RNA isolated from *A. flavus*-infected cotyledons revealed a significant reduction of targeted *nsdC*, *veA*, *aflM*, and *aflR* transcripts, indicating silencing of the targeted fungal genes during the *A. flavus* infection of HIGS groundnut cotyledons, compared to their WT counterparts (Figure 1E).

### 2.2. 4RNAi-Expressing HIGS Lines Demonstrate no Substantial Alterations in Gene Expression

Any unintentional suppression of nontargeted groundnut genes was assessed by predicting siRNA sequences using pssRNAit for the 4RNAi gene sequences [22]. The predicted siRNAs revealed no putative off-targets in the groundnut genome (Appendix A). Moreover, HIGS plants did not show any noticeable effects on their growth and development and demonstrated normal morphology, flowering, and seed set compared to their WT controls (Appendix A).

### 2.3. HIGS Cotyledons Showed Significant Resistance to A. flavus Infection and Aflatoxin Contamination

The cotyledons of T_1,_ T_2,_ and T_3_ progenies of groundnut HIGS lines were screened for *Aspergillus* infection using in vitro seed colonization assays. Of these, progenies of two HIGS lines (B-10-7 and F-5-4) consistently had less mycelial growth compared to WT controls after challenging with the *A. flavus* strain AF11-4 (Figure 2A, Appendix A). Furthermore, the relative gene expression of the *FLAV* gene significantly varied between WT control lines and 4RNAi events. The 4RNAi event B-10-7 and F-5-4 showed a 97% and 99% reduction, respectively, in fungal biomass compared to the susceptible WT control line (Figure 2B).

Similarly, the level of aflatoxin B1 tested across 7 T1 events showed a significant (*p* ≤ 0.01) reduction of aflatoxin levels in the inoculated 4RNAi cotyledons (0–6 ppb) compared to the WT-controls (7529.27 ppb) (Figure 2C). Trait stability was confirmed in the T_2_ and T_3_ generations, while the AFB1 levels were significantly reduced in the T_2_ (0–3 ppb) and T_3_ (0.1–17 ppb) progenies, compared to the WT controls (Figure 2C).

### 2.4. Impact of Aspergillus Infection on Groundnut Proteomes

To understand the proteome changes for identifying *A. flavus*-responsive proteins, label-free quantitative proteomics analysis was performed on *A. flavus*-infected HIGS and WT lines at 0, 30, 48, and 72 h post-infection. Proteins were identified based on the criteria of at least 2 unique peptides matching with a 1.5-fold change at *p* ≤ 0.05 between the uninfected and infected groups in the contrasting genotypes and expression profiles studied (Figure 3A). Comparative analysis revealed differential regulation of 984 proteins at various time points, with 528 proteins being upregulated and 456 downregulated in HIGS lines compared to the WT controls (Appendix A).

Based on gene ontology analysis, the identified proteins were categorized into different groups, such as molecular functions, cellular components, and biological processes (Appendix A). The data of GO-based annotation, KEGG pathways, and the subcellular localization of all identified proteins are listed in Appendix A. Putative function analysis identified 21 GO terms related to biological processes, 9 GO terms for molecular functions, and 6 GO terms for cellular components. Proteins with binding and catalytic activity were highly represented in the molecular function category, while the cells, cell parts, and organelles were the most represented categories of cellular components. Most proteins were involved in cellular processes, response to stimulus, and metabolic processes in the biological process category. DEPs in the contrasting lines shared the same categories in biological, molecular, and cellular processes in broad functional distribution analysis. However, there were differences in the proportional distribution of the proteins (Appendix A).

Pathway analysis carried out between DEPs from HIGS and WT control samples using the MapMan tool revealed groundnut metabolic Aspergillus pathways that responded to *A. flavus* infection. The DEPs were mapped individually and in comparison to each other and different functional categories. All DEPs were associated with 35 pathways (Appendix A), with significantly enriched ones related to stress (24–26%), signaling (9–11%), protein (9–10%), lipid metabolism (6–7%), and photosynthesis (5%).

### 2.5. Effect of Host-Induced Gene Silencing on the A. flavus Proteome

To confirm whether the inhibition of aflatoxin biosynthesis observed in our study occurred through host-induced gene silencing of *nsdC*, *veA*, *aflM*, and *aflR*, the proteomic analysis of the HIGS lines and WT control were compared under *A. flavus* infection. Proteins were filtered against the known *A. flavus* proteome on the UniProt database. A total of 1745 DEPs were observed, of which 995 were upregulated and 750 were downregulated in the HIGS lines in comparison with the WT controls (Appendix A), and expression profiles of the quantified proteins were heat-mapped (Figure 3B, Appendix A).

Proteins previously reported to play a role in fungal differentiation and development, pathogenicity, and aflatoxin biosynthetic pathways were selected, and their expression was compared based on the fold change (FC) between the 4RNAi and WT samples (Table 1, Figure 3C). Genes known to be involved in fungal differentiation and pathogenicity—such as calmodulin (46.44 FC); eukaryotic translation initiation factor 3 subunit I (35.84); transcriptional activator *hacA* (23.18); kynurenine 3-monooxygenase 2 (11.03); fungal sexual development regulator *velC* (3.47 FC); conidiophore development regulator *veA* (2.44 FC); and aflatoxin biosynthetic pathway proteins, such as *aflC* (7.06 FC), *aflL* (3.79 FC), *aflM* (4.26 FC), *aflQ* (3.46 FC), *aflR* (2.01 FC), *aflS* (5.71 FC), *aflV* (3.92 FC), *aflW* (1.75), and *aflJ* (1.90)—were higher in the WT lines, while pectinesterase A (77.76), pectin lyase D (21.05), calpain-7 (12.10), *aflB* (2.16 FC), *aflN* (2.74), and *α-amylase A* (2.82 FC) were in higher abundance in HIGS samples.

The levels of DEPs observed in the proteomics data of the known aflatoxin biosynthetic pathway proteins were further validated by qRT-PCR analysis of their respective genes (Figure 4). The results were consistent with the proteomics data, confirming the downregulation of proteins upon the silencing of the targeted aflatoxin regulatory and cluster biosynthetic genes. Expression of the targeted developmental and aflatoxin regulatory genes was significantly reduced compared to the WT control samples in the two promising lines, viz., 4RNAi_B and 4RNAi_F, for *nsdC* (0.70 and 0.75 FC), *veA* (0.73 and 0.51 FC), and *aflR* (0.11 and 0.24 FC), as well as the biosynthetic genes *aflM* (0.35 and 0.27 FC), *aflQ* (0.05 and 0.66 FC), *aflV* (0.05 and 0.72 FC), *aflW* (0.04 and 0.49 FC), *aflS* (0.19 and 0.77 FC), *aflL* (0.16 and 0.66 FC), *aflB* (0.05 and 0.53 FC), *aflP* (0.19 and 0.56 FC), *aflD* (0.04 and 0.50 FC), *aflO* (0.05 and 0.74), and *aflJ* (0.14 and 0.43 FC).

### 2.6. Differentially Expressed Proteins in the Groundnut Host System and Identification of Host Resistance-Associated Proteins

To identify the potential proteins that are associated with resistance, proteomes of the HIGS lines were compared to the WT line during the progression of *A. flavus* infection (Table 2, Appendix A). Analysis of *A. flavus*-responsive DEPs revealed that they were essentially involved in the activation of heat shock proteins (HSPs), calcium signaling, phytohormones, transcription factors, and fatty acid pathways (Table 2, Figure 5).

Levels of resistance-associated proteins—including Ca^2+^ signaling proteins, such as calcium-dependent protein kinase (CDPK) (5.94 FC) and Ca^2+^ binding protein (SOS3) (2.27 FC)—were not detected or significantly higher in HIGS lines compared to the WT control lines at different time points, except for CDPK that was downregulated at the 0 and 30 h time points. Further, levels of Ca^2+^ signaling proteins involved in heat shock signal transduction-activated heat shock proteins, such as HSP17.6 (2.50 FC), HSP 2 (2.50 FC), HSP 70 (3.62 FC), and HSP transcription factor A-2 (2.10 FC), were not detected or significantly higher in the HIGS samples compared to the control at most time points.

Proteins encoding phytohormone and transcription factors were detected in abundance in HIGS lines compared to the WT control samples (Table 2). Phytohormones, such as auxin-induced putative Aldo/keto reductase family protein (4.44 FC), auxin signaling F-box 3 (3.38 FC), ABA response element-binding protein 1 (12.57 FC), and ABA 8′-hydroxylase 3 (8.95 FC), were either not detected or significantly higher in HIGS lines than in the WT control. Similarly, HIGS lines showed significant upregulation of several transcription factors, including ethylene-responsive transcription factor (66.16 FC), ethylene-responsive element-binding factor 3 (70.87 FC), ethylene-responsive element-binding factor 4 (6.26 FC), NAC (4.04 FC), NAC3 (4.28 FC), MYB25 (5.33 FC), MYB1 (4.73 FC), MYB9 (2.40 FC), WRKY15 (12.15 FC), and DREB transcription factor (9.65), except for WRKY, which was not detected at 0 and 72 h and was downregulated at 30 h.

Numerous DEPs were associated with different metabolic pathways in the HIGS lines compared with the susceptible WT control plants (Table 2). Some of the proteins that changed substantially were related to fatty acid biosynthesis, which includes acyl carrier protein (29.37 FC), lipoxygenase (9.88 FC), lipoxygenase 1 (7.97 FC), phosphatidylinositol phosphate kinase (4.68 FC), lysophosphatidyl acyltransferase 5 (5.40 FC), palmitoyl-monogalactosyldiacylglycerol Δ-7 desaturase (6.01 FC), ceramide kinase-related protein (3.18 FC), sphingolipid Δ-8 desaturase (9.71 FC), and phospholipase D (4.15 FC), which were higher in HIGS lines compared to the control line.

### 2.7. Identified Host Susceptibility-Associated Proteins

Higher levels of susceptibility-associated proteins (SAPs) were observed in WT when compared to HIGS lines (Table 3, Appendix A). Significantly higher levels of putative SAPs were detected for proteins—such as annexin (6.26 FC); syntaxin (4.02 FC); mildew resistance locus O (MLO)-like protein (3.82 FC); calmodulin (3.33); heat shock protein HSP4 (11.67 FC); transcription factors, such as NAC 2 (5.56), MYB 21 (5.59), and MYB 20 (3.00); β -ketoacyl-ACP synthase II-1 (41.34 FC); 9-cis-epoxy carotenoid dioxygenase (7.53); long-chain acyl-CoA synthetase 4 (12.32); and C3HC4 type (RING finger) (7.70 FC)—than for the WT lines during *A. flavus* infection (Table 3).

### 2.8. Validation of DEPs by qRT-PCR

Selected DEPs identified through proteomics were further validated by qRT-PCR of their associated genes to ascertain if changes observed in protein expression were regulated during transcription (Figure 6). These were previously reported to be involved in various biological processes in response to biotic stress, proteolysis, flavonoid and fatty acid biosynthesis, and signal transduction pathways and were grouped into different subcategories that are linked to plant resistance or susceptibility, either directly or indirectly. The fold change in the expression of target resistance genes in the two HIGS lines compared to the WT control following *A. flavus* infection was validated by qRT-PCR. Significant increases in expression (*p* ≤ 0.05) were observed in both HIGS samples for calcium-dependent protein kinase (CDPK) (4.89, 3.72 FC), cinnamyl alcohol dehydrogenase (CAD) (3.39, 4.47 FC), cinnamic acid 4-hydroxylase (C4H) (3.17, 7.28 FC), chalcone-flavanone isomerase (CFI) (2.44, 5.25 FC), cationic peroxidase 2 (PNC) (7.04, 72.62 FC), diacylglycerol acyltransferase (10.5, 22.39 FC), dihydroflavonol-4-reductase (DFR) (1.62, 4.46 FC), lysophosphatidyl acyltransferase 5 (LPAT) (1.26, 7.29 FC), sphingolipid Δ-8 desaturase (SLD) (1.83, 15.74 FC), and calmodulin (5.03, 1.53 FC)). These results were consistent with the proteomics data, confirming the differential expression of all 10 genes after infection by the pathogen in the resistant HIGS lines when compared to the WT control line.

## 3. Discussion

Host-induced gene silencing of *A. flavus* genes essential for pathogen growth and development has proven to control both necrotrophic and biotrophic fungal pathogens [23,24,25]. Several studies have reported that upon *A. flavus* infection, selective degradation of mRNA induced by siRNA interferes or blocks the translation of the targeted fungal genes, resulting in reduced aflatoxin contamination [8,9,10,14,17].

The HIGS groundnut plants developed in this study simultaneously target *nsdC*, *veA*, *aflM*, and *aflR* involved in fungal morphogenesis and aflatoxin biosynthesis pathway genes [18,26], and the HIGS plants demonstrated significantly lower infection and aflatoxin accumulation than previously reported in other RNAi-based studies [8,9,14]. PCR and RT-PCR screening showed stable expression and inheritance of the 4RNAi construct in progenies from the T_2_ and T_3_ generations. Segregation analysis revealed mendelian segregation of transgenes, thereby indicating inheritance of single copy inserts in a 3:1 ratio, aligning with our previous report [9]. Our results confirm previous findings, where silencing of the fungal sexual development gene, *nsdC*, in *A. flavus* demonstrated a lower fungal load and aflatoxin production [18]. After downregulation of *aflR*, a regulatory gene, a sequence-specific zinc, binuclear, DNA-binding protein that activates the transcription of most structural genes in the aflatoxin gene cluster was shown to suppress the expression of *A. flavus* pathway genes [27,28]. Silencing of the *aflR* gene was previously shown to result in significantly lower levels of aflatoxins (14-fold) in RNAi maize plants than in wild-type plants, though significant off-target effects on plant architecture were also observed [13]. Another candidate for our study was *veA*, a velvet family protein that plays a key role in *A. flavus* conidiation and sclerotial, as well as regulating aflatoxin biosynthesis [29]. The downregulation of *veA* suppresses the expression of *aflR*, *aflD*, *aflM*, and *aflP*, the major aflatoxin genes, resulting in inhibition of aflatoxin synthesis in the fungus [30]. Likewise, RNAi-based suppression of another target gene, *aflM*, was previously shown to significantly enhance aflatoxin resistance in maize [14], and RNAi groundnuts [9] also demonstrated significantly enhanced resistance to aflatoxin contamination.

Several HIGS lines developed in this study significantly reduced the *A. flavus* biomass compared to WT control lines, which could be attributed to the silencing of the targeted *nsdC* and *veA* genes. These lines also showed high levels of aflatoxin resistance, with the HIGS lines accumulating non-detectable levels (<10 ppb) of aflatoxin in comparison to >7000 ppb in WT lines. Gene expression studies indicated over 50% reduction of the transcripts of the fungal genes *aflM* and *aflR* in the 4RNAi-HIGS lines assayed, whereas *nsdC* and *veA* showed 30% suppression in the tested HIGS lines compared to WT controls during infection. This showed that *aflM* and *aflR*, and to a lesser extent *nsdC* and *veA*, are limiting factors in aflatoxin biosynthesis and are efficient targets for HIGS, as reported in maize [13,14].

Comparative proteome profiling in the HIGS lines and their WT counterparts gathered further evidence that suppression of these four RNAi-targeted genes affected the fungal morphogenesis and aflatoxin cluster genes. A significant reduction in the expression of several fungal proteins in the infected HIGS lines was observed, including the fungal sexual development regulator, VelC, and aflatoxin biosynthetic pathway proteins, such as AflC, AflL, AflM, AflQ, AflR, AflS, AflV, AflW, VeA, and AflJ, further validating the reduced growth of the fungus on the HIGS groundnut lines, as was demonstrated in the bioassays. Silencing of five genes involved in aflatoxin production, *aflR*, *aflS*, *aflC*, *pes1*, and efflux pump (*aflep)*, has been previously reported in groundnut to result in a 100% reduction in the aflatoxins B1 and B2 [8]; however, this study reported lower levels of aflatoxin only in immature seeds [31]. Several in vitro studies also revealed that RNAi-based silencing of aflatoxin pathway genes causes a significant reduction in aflatoxin production [32,33]. These findings suggest that simultaneous silencing of morphogenesis and aflatoxin cluster genes can be an attractive strategy for reducing aflatoxin content in groundnut.

Despite the demonstrated success of HIGS as an effective aflatoxin mitigation strategy, the molecular mechanisms of resistance to *Aspergillus* infection and aflatoxin contamination in plants is not well understood. Hence, we compared proteome profiles of HIGS lines expressing the 4RNAi construct and WT controls during *A. flavus* infection. We identified differential expression of resistance-associated proteins or susceptibility-associated proteins during the groundnut–*A. flavus* interaction. Intrinsically, plants have different barriers to prevent the entry and growth of the pathogen, including the cell wall, which plays a significant role [34]. When the core defense mechanism of plants is ineffective, they begin to rely on the gene products that can recognize and respond to pathogen effector molecules, known as host plant effector-triggered immunity (ETI). At the molecular level, the interaction between plant and pathogen is a mutual interplay, where calcium signaling pathways either activate or deactivate the ROS pathway [35]. We observed high expression of calcium-dependent protein kinase, SOS3 proteins, HSP17.6, HSP70, HSP2, and heat shock transcription factor A-2 in the resistant HIGS lines. The high concentrations of calcium ions in the cytosol affect the production of enzymes that generate reactive oxygen species (ROS) [35,36], which further regulate the heat shock proteins (HSPs) in pathogen infection as defense molecules. Furthermore, the increased levels of free calcium can activate the mitogen-activated protein kinases (MAPKs), which play a key role in the phosphorylating of regulatory proteins. Heat shock proteins (HSPs) function as molecular chaperones by interacting with other proteins and providing stability and protection from damage [37]. For instance, HSP, Ntshsp17, and RSI2 act as molecular chaperones and help in inducing defense responses in tobacco and tomato against *Ralstonia solanacearum* and *Fusarium oxysporum*, respectively, by stabilizing signaling-related proteins [38]. Likewise, in tomatoes, the induction of mitochondrial HSP22 during oxidative stress helps to provide adaptive responses [39]. In contrast, Mds1 (Mayetiola destructor susceptibility-1) expression in wheat leads to an increased susceptibility to wheat gall midge and powdery mildew [40]. In addition, several reports suggest that ROS also induces an increase in cytosolic Ca^2+^ concentrations, which in turn activates other defense responses, such as the production of phytohormones, transcription factors, and secondary metabolites [41,42,43].

We observed that the proteins associated with the phytohormones synthesis, including auxin, gibberellin, ethylene, and ABA, that is involved in host–pathogen interactions were induced at higher levels in resistant HIGS groundnut lines [44]. ABR1, a homolog of an abscisic acid insensitive gene, which is known to be a repressor of the ABA signaling pathway, has been reported to confer resistance against pre-harvest aflatoxin contamination [45]. Further, in maize, the ethylene-responsive factor (ZmERF1) was shown to induce defensin proteins that resist *Aspergillus* infection [46].

Over 40 transcription factor-related proteins, including MYB, WRKY, NAC, and ERF binding proteins, were detected in the resistant HIGS lines at high levels. Among these was a transcription factor, MYB30, that is a positive regulator of a hypersensitive response (HR) involved in the regulation of downstream very-long-chain fatty acid (VLCFA) biosynthesis pathways in *Arabidopsis* against pathogen attack [47]. Furthermore, Apple MdMYB30 has been shown to modulate plant resistance by regulating cuticular wax biosynthesis against *Botryosphaeria dothidea* [48]. In addition, WRKY genes are reported to regulate fatty acid composition in *Arabidopsis* [49] and positively influence the PR1 protein activity in rice [50] during *Xanthomonas oryzae* pv. *oryzae* (*Xoo*) attacks, and they are involved in defense responses to *A. flavus* inoculation in maize [46].

Plants have developed specific metabolic pathways to synthesize signaling molecules and antimicrobial compounds to combat pathogen infection. In the current study, 52 proteins were differentially induced in the resistant HIGS lines that are involved in fatty acid metabolism. We observed a higher abundance of acyl carrier protein, lipoxygenase, β-hydroxy acyl-ACP dehydratase, phosphatidylinositol phosphate kinase, lysophosphatidyl acyltransferase 5, palmitoyl-monogalactosyldiacylglycerol Δ-7 desaturase, ceramide kinase-related protein, sphingolipid Δ-8 desaturase, and phospholipase D in 4RNAi lines. The antimicrobial properties of plant lipoxygenases were reported for various pathogens, including *A. flavus* [51]. In maize and soybean, lipoxygenase-3 (LOX3) and a few other 9-oxylipins suppress aflatoxin biosynthesis upon *A. flavus* infection [52]. The glycerophospholipids are structural components of membranes that act as novel secondary messengers as defense signaling pathways in plants [53]. Similarly, Phospholipase D (PLD) catalyzes the hydrolysis of structural phospholipids functioning as second messengers in the regulation of signaling pathways in plant defense [54]. Expression of the α, β, and γ class of *Phospholipase* genes is induced following *Pseudomonas syringae* infiltration in *Arabidopsis*, thereby suggesting their function as a positive regulator of disease resistance [55]. Hence, this provides insights into the involvement of fatty acids in the synthesis of signaling molecules and antimicrobial compounds to act as physical and chemical barriers to the entry of *Aspergillus* during the infection process [41,56].

Furthermore, we observed that the susceptibility-associated proteins (SAPs), such as mildew resistance locus O (*MLO*), annexin, syntaxin, calmodulin, and 9-cis-epoxy carotenoid dioxygenase, were significantly upregulated in susceptible WT controls compared to HIGS lines. Understanding the role of these susceptibility genes helps us to devise strategies for breeding aflatoxin-resistant crops [41,57]. The primary calcium sensor in plants, calmodulin (CaM) binds to calcium ions and regulates various cellular functions by modulating the activity of different target proteins in response to calcium signals [58]. Silencing of the calmodulin-like proteins *SlCML55* in tomatoes inhibits Phytophthora infection [59]. The 9-cis-epoxy carotenoid dioxygenase involved in the biosynthesis of ABA was reported to be highly expressed in the susceptible genotype. Increased ABA levels suppress disease resistance by downregulating salicylic acid (SA)- or methyl jasmonate (MJ)-induced defense gene expression [60]. However, depending on the pathosystem, ABA reportedly modulates host immunity against fungal pathogens. In the rubber plant, ABA has been shown to positively regulate the defense against powdery mildew [61], and in barley, ABA contributed to an increased susceptibility to *M. oryzae* [62]. In addition, as reported previously [41], genes such as MLO, annexin, and syntaxin were also identified as SAPs in this study. Since *MLO* helps in fungus attachment to the host cell, increasing host susceptibility (S) to fungal pathogens, silencing of these genes enhanced disease resistance against the powdery mildew in different crops, indicating its negative role in plant defense [63]. Annexins, a family of calcium-binding proteins that mediate membrane fusion and regulate the phagocytosis and exocytosis of vesicles [64], were upregulated in the WT and possibly might have decreased the integrity of the plant cell membrane, resulting in increased susceptibility. Similarly, the upregulation of syntaxin during infection indicated its role as a susceptibility factor that promotes infection. RNAi-mediated silencing of syntaxin has been reported to confer resistance to *P. infestans* in potatoes [65] and apples [66]. Considering these findings, our results justify further study of these SAPs as potential targets in gene editing approaches for enhanced resistance to aflatoxin contamination in groundnut.

## 4. Conclusions

We propose an effective mechanism to alleviate aflatoxins in groundnuts by reducing their levels quite effectively below the regulatory thresholds. Our study provides ample evidence that groundnut can export heterogenous expressed sRNAs into the invading fungus, and that silencing of target genes that are essential for pathogen growth, development, and aflatoxin production affect the pathogenicity and resulting mycotoxin accumulation in *Aspergillus*–groundnut pathosystems. Furthermore, comparative proteome profiling of the HIGS lines during infection provided clues that targeting the fungal *nsdC* and *veA* genes could have allowed the HIGS genotype more time to mount a strong defense response to the invading fungus compared to the WT host, resulting in higher levels of resistance gene expression triggering a natural defense mechanism, whereas simultaneous suppression of *aflR* and *aflM* genes disrupted the aflatoxin biosynthetic pathway, resulting in reduced contamination. Together, silencing multiple *Aspergillus* genes by HIGS conferred enhanced resistance, providing an effective strategy for controlling fungal infection and aflatoxin contamination. A major highlight of this work is the identification of genes and their encoded proteins that play a role in the complex innate defense mechanisms of groundnut and perhaps other plant species that can serve as key molecular targets in future metabolic engineering or breeding approaches for developing aflatoxin-resistant crops.

## 5. Materials and Methods

### 5.1. Generation of HIGS Groundnut Expressing RNAi Genes

Four *A. flavus* genes, including the fungal developmental genes *nsdC* (GenBank: CP044620.144), *veA* (XM_041294274.1) and the aflatoxin biosynthetic pathway genes *aflR* (XM_041285628.1) and *aflM (ver-1)* (XM_041291516.1), were isolated from a highly aflatoxigenic *A. flavus* isolate NRRL 3357 [67]. PCR products of four regions of the *A. flavus* genes *nsdC* (210 bp, CP044620.1:2938046–2938256), *veA* (200 bp, XM_041294274.1:1171–1371), *aflM* (210 bp, XM_041291516.1: 2152–2362), and *aflR* (200 bp, XM_041285628.1:840–1040) were cloned into the pHANNIBAL vector downstream to the cauliflower mosaic virus (CaMV) 35S promoter in sense and antisense orientations, separated by a pyruvate orthophosphate dikinase (Pdk) intron with a polyadenylation signal. To avoid the off-target effects, siRNAs for the 4RNAi gene sequences were predicted using pssRNAit [22]. These regions having efficient siRNA hits showing no homology with the sequence of the groundnut genome were selected for construct development (Appendix A). The resulting recombinant gene (2x35S-5′4RNAi-Pdk-3′4RNAi-polyA) was then cloned at *Eco*RI and *Hind*III sites of the binary vector pPZP200; this construct is referred to as pPZP200_4RNAi throughout the text (Figure 1A,B). The binary vector pPZP200 contains a spectinomycin gene for bacterial selection and is devoid of a selectable marker/reporter gene. HIGS lines were developed in groundnut var. ICGV 91,114, using *Agrobacterium*-mediated transformation [68,69]. Regenerated plants were transplanted into soil and grown to maturity under containment greenhouse facilities.

### 5.2. Genotyping of HIGS Plants

The DNeasy^®^ Plant Mini Kit (Qiagen Valencia, Valencia, CA, USA) was used to isolate the genomic DNA from young leaves of putative groundnut transformants. Extracted DNA was quantified using a Qubit™ 4 Fluorometer (Invitrogen, Waltham, MA, USA). The presence or absence of the 4RNAi transgenes and a control gene, *ADH 3* (EG529529), were determined using PCR with gene-specific primers (Appendix A). PCR was performed using Emerald Amp^®^ GT PCR 2× Master Mix (Takara Bio Inc., San Jose, CA, USA) as per the manufacturer’s instructions.

### 5.3. Gene Expression Analyses

Leaf tissues were used to extract total RNA for RT-PCR, while infected kernel tissues collected at 0 and 48 h post-infection (hpi) were used for qPCR. The RNeasy Plant Mini kit (Qiagen, Tokyo, Japan) was used for RNA extraction and quantified with a Nanodrop ND 1000 Spectrophotometer (Nanodrop, Wilmington, DE, USA). DNase treatment was carried out using the DNase Kit (Fermentas, Waltham, MA, USA) according to the manufacturer’s instructions. For cDNA synthesis, 1.0 μg of purified RNA was used with the recommended protocol (Thermoscript RT-PCR system, Invitrogen, Waltham, MA, USA).

RT-PCR was performed using an Emerald Amp^®^ GT PCR 2× MasterMix (Takara Bio Inc., San Jose, CA, USA), as per the manufacturer’s instructions, followed by gel electrophoresis to visualize PCR products. Quantitative PCR was performed in a Realplex Real-Time PCR system (Eppendorf, Framingham, MA, USA) using SYBER Green mix (Bioline, London, UK). For gene expression analysis, the groundnut *ADH 3* (EG529529) and *G6Pd* (EG030635) genes were used as reference genes [70], while the *A. flavus* species-specific tubulin (CP044619.1) was used as the reference gene for fungal gene expression (Appendix A). Fungal gene primers for qPCR were designed outside of the sequence used to obtain ds4RNAi.

For each qPCR reaction (10 µL), 5-times-diluted cDNA, 0.4 mM of each primer, and 5 µL SYBR Green mix (Bioline, London, UK) were added. The reaction conditions involved initial denaturation at 95 °C for 2 min, followed by 40 cycles of 15 s at 95 °C and 30 s at 60 °C with fluorescent signal recording. Melt curves were generated for each reaction to ensure specific amplification. All qPCR reactions, including the non-template control, were performed in 3 biological and 3 technical replicates. Relative fold expression was determined using the 2^−ΔΔCt^ method [71].

### 5.4. Copy Number Detection and Inheritance Studies

To estimate the copy number in the 4RNAi lines, a relative quantitative method [72] was considered using the equation: X_0_/R_0_ = 10^[(^**^Ct^_X_^I^_X_^)/S^_X_^][(Ct^_R_^-I^_R_^)/S^_R_^]^**, where I_X_ and I_R_ represent intercepts of the relative standard curves, and S_X_ and S_R_ represents slopes of the standard curves of the target and reference genes, respectively. Ct_X_ and Ct_R_ are the threshold cycles for amplification of the target and reference genes of the unknown test sample. The serial dilutions of genomic DNA from 100 ng to 10 ng were used to prepare the standard curves for the transgene (*aflR*), and vacuolar protein sorting-associated protein 53 A-like (*GnVP*) was used as the endogenous/reference gene. Copy numbers were detected by qPCR in the Realplex (Eppendorf, Hamburg, Germany) Real-Time PCR system using a 2X SensiFAST^TM^ SYBR No-ROX (Bioline, London, UK) kit with 10 ng of genomic DNA for each sample (using three biological and technical replicates for each event). Standard curves were generated by plotting the log values and the corresponding Ct values. The copy number of the reference gene (R_0_) for *GnVp* (Gene ID: 107638771) was taken as 2 copies in the tetraploid groundnut for copy number estimation. Primer sequences for target and reference genes are given in Appendix A. PCR data of T_1_, T_2,_ and T_3_ generation were used to analyze the segregation pattern in the 4RNAi events. The Chi-square test (*p* < 0.05) was applied to determine if the 4RNAi transgene was segregating according to the Mendelian pattern of inheritance.

### 5.5. Aspergillus flavus Growth Conditions

Fungal bioassays were conducted using the AF 11-4 strain of *Aspergillus flavus* [73]. Fungal spores were collected in sterile distilled water containing 0.05% Tween 20 and diluted to a concentration of 5 × 10^4^ spores/mL using a Neubauer hemocytometer. The colony-forming units (CFUs) were determined by standard tenfold dilutions to obtain ~40,000 CFU/mL on *A. flavus parasiticus* agar (AFPA) medium.

### 5.6. Colonization of A. flavus on Kernels and Aflatoxin Analysis

The T_1_, T_2_, and T_3_ generation HIGS lines were screened for resistance to a highly aggressive and toxigenic strain of *A. flavus* isolate AF 11–4 by in vitro seed colonization, as previously described [9]. Seeds from the HIGS events and WT control ICGV 91,114 were surface sterilized with a 0.1% solution of mercuric chloride, followed by washings with sterilized distilled water. The seeds were soaked in water for 2 h, and subsequently, the seed coat was removed, and cotyledons surgically cut into two vertical halves and arranged with the cut surface exposed in Petri dishes with 1.7% sterile agar/water (*w/v*). Each half of the cotyledon was inoculated with 2 µL freshly prepared fungal spore suspension (5 × 10^4^ spores/mL) and incubated at 28 ± 1 °C in the dark. All the inoculated and uninoculated samples collected at 0, 30, 48, and 72 h post-infection (hpi) were immediately frozen in liquid nitrogen and stored at −80 °C for further use.

The aflatoxin content (AFB1) was estimated using the indirect ELISA method, as previously described [74]. Briefly, the cotyledons collected at 72 hpi were extracted in 70% methanol containing 0.5% KCl and incubated overnight for 16 h in the dark at 25 °C, followed by ELISA. Three biological and three technical replicates were used in aflatoxin bioassays, along with wild-type controls (cv. ICGV 91114). The biological replicate consisted of 100 mg of each half-cotyledon sample per event.

### 5.7. Detection of Fungal Load

The DNA was extracted from 100 mg of *A. flavus*-infected groundnut seeds at 72 hpi with the PureLink Plant Total DNA Purification kit (Invitrogen, Waltham, MA, USA). The isolated DNA was evaluated for purity on 0.8% (*w/v*) agarose gel, and the concentration was determined using a Qubit Fluorometer 2.0 and stored at −20 °C for further use. The fungal load in the *A. flavus*-infected groundnut samples of the WT control and 4RNAi lines was determined using quantitative PCR (qPCR) with a relative quantification method [75]. The DNA concentration of each sample was normalized to 50 ng/µL. Following the test for DNA amplification using groundnut *ADH 3* (EG529529) gene-specific primers, quantitative real-time PCR (qPCR) was performed to amplify the *A. flavus* ITS2 region, using *FLAV* as the target gene and *ADH3* as the housekeeping gene (primer sequences shown in Appendix A). The qPCR reaction (10 µL) included 1 µL of template DNA, 0.4 mM of each primer, and 5 µL SYBR Green mix (Bioline, London, UK). The qPCR reactions were performed in biological and technical triplicates, and the Ct values for the *FLAV* gene were normalized using the groundnut *ADH3* gene. The relative gene expression of *FLAV* was calculated using the 2^−ΔΔCt^ method [71].

### 5.8. Extraction and Digestion of Total Proteins

Proteins were extracted from both *A. flavus*-infected and uninfected samples of T_3_ generation HIGS lines (F-4 & F-5) and the WT control ICGV 91114, as previously described [41]. The protein concentration was determined and normalized by loading an equal amount of each sample in the polyacrylamide gel electrophoresis (PAGE). The proteins were then subjected to reduction, alkylation, and overnight trypsin digestion using sequencing-grade porcine trypsin (Promega, Madison, WI, USA). Peptides from each fraction were extracted separately in 60% (*v/v*) acetonitrile (ACN) containing 0.1% (*v/v*) formic acid, sonicated in ice for 30 min, followed by concentrating in a speed vacuum concentrator (Thermo Scientific, Waltham, MA, USA) and purification using C18 spin columns (Thermo Scientific, Waltham, MA, USA). These samples were either immediately used for proteomics analysis or stored at −80 °C for further use.

### 5.9. UPLC-MS Analysis of Peptides

The trypsin digests were separated on an Acquity BEH C18 UPLC column (75 µm × 150 cm × 1.7 µm; Waters, Cheshire, UK) connected to a UPLC system for 90 min using LC-MS grade water in 0.1% formic acid (v/v; mobile phase A) and acetonitrile in 0.1% formic acid (mobile phase B). The separated peptides were analyzed for MS and MS/MS fragmentation on a Xevo-G2-XS (Waters, Milford, MA, USA), using an ESI source in positive mode. The scan time was set to 0.5 s in continuum mode, and the mass range was set to 50–2000 Da in TOF, with Leucine encephalin (200 pg/μL; Sigma-Aldrich, St. Louis, MO, USA) as an external calibrant. Mass spectra of the samples were acquired by MassLynx v4.0 software (Waters, Milford, MA, USA) and submitted for the identification and expression analysis of proteins.

### 5.10. Identification and Quantification of Proteins

Progenesis QI for Proteomics Software V.4.0 (Nonlinear Dynamics, Newcastle, UK) was used to identify and quantify proteins from raw spectra. Following alignment of the MS scans by retention time, peak lists for each replicate containing *m/z* and abundance were generated. Parameters were set to 5 to increase sensitivity for peak detection, and proteins with charges higher than 5 in the detection range (100–1600 *m*/*z*) between 5–80 min were captured. Triplicate MS scans grouped based on the treatment were normalized and compared for abundance data, followed by statistical evaluation. The peptides assigned to more than one protein were excluded from quantification. Protein identifications were completed using the Progenesis inbuilt ion accounting algorithm against the groundnut proteome and *A. flavus* proteome downloaded from UniProt (http://www.uniprot.org/, accessed on 30 September 2021), where the false discovery rate (FDR) for statistically significant proteins was 5% [76]. Quantitative analysis was based on the ratio of protein ion counts among contrasting samples or treatments. The resulting dataset was filtered, and only proteins quantified with 2 or more unique peptides and having a fold change of 1.5 with a significant *p*-value (*p* ≤ 0.05) were considered as up- or downregulated. Significant protein subcellular localization was predicted using Plant-mPLoc [77]. The proteomics data based on mass spectrometry were deposited at the ProteomeXchange consortium via jPOSTrepo [78] with the database identifier PXD028196.

### 5.11. Functional Annotation and Pathway Mapping

The protein classification analysis of unique and differentially expressed proteins into Gene Ontology (GO) categories was conducted using the MapMan application (http://mapman.gabipd.org, accessed on 30 September 2021) [79]. The potential involvement of these predicted proteins in biological pathways was explored by mapping them to the reference canonical pathways, using the *Arachis* genus as a reference in the KEGG automated annotation server KAAS (http://www.genome.jp/kaas-bin/kaasmain, accessed on 30 September 2021). To identify the expression patterns of proteins responsive to *A. flavus*, 50 common proteins that showed equal to or more than 1.5-fold variation were hierarchically clustered, using MeV software (Version 4.8, USA) with the Euclidean distance method.

## Figures and Tables

**Figure 1 toxins-15-00319-f001:**
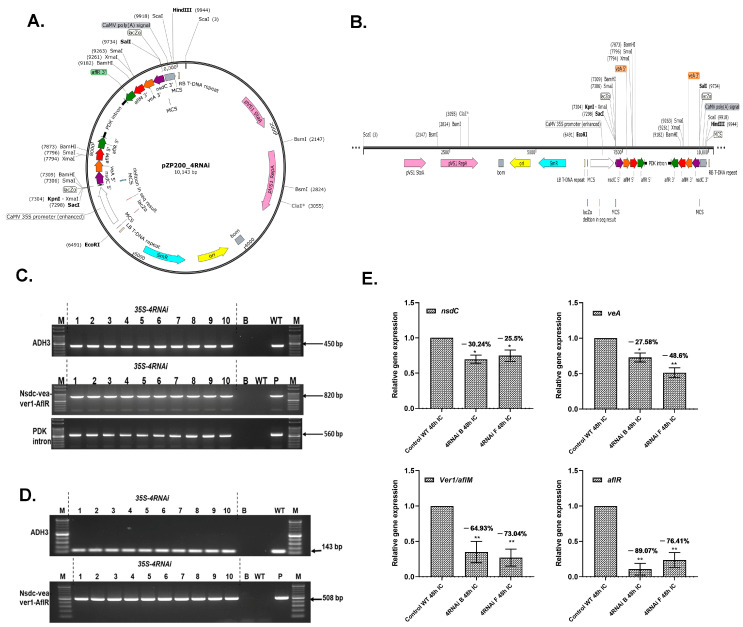
Development of groundnut 4RNAi-HIGS lines. (**A**) Circular map of the 4RNAi binary vector used for groundnut transformation. (**B**) Linear representation of T-DNA region of the 4RNAi binary vector containing the constitutive enhanced cauliflower mosaic virus (d35S CaMV) promoter used for targeting the aflatoxin pathway genes. The hpRNA cassettes have inverted repeats of respective *nsdC*, *veA*, *aflM*, and *aflR* gene regions highlighted in purple, orange, red and green color respectively under control of the d35S CaMV promoter. LB left border; RB, right border. (**C**) PCR analysis using the 4RNAi primer pair to confirm the presence of 4RNAi genes (820 bp) and the primer pair specific for the PDK intron. (**D**) RT-PCR analysis of 4RNAi events. cDNA from HIGS and WT control were used to amplify the inserted transgene with amplicon sizes of 508 bp using the 4RNAi2 primer pair and primers specific for the endogenous gene (ADH3), (**E**) Relative transcript expression of *A. flavus nsdC*, *veA*, *aflR*, and *aflM* from infected 4RNAi lines and the WT control line. Quantitative RT-PCR of RNAs isolated from 48 hpi (hours post infection) samples used the *A. flavus* housekeeping gene, beta-tubulin, as the normalizer. Significant differences between HIGS and wild-type control plants were analyzed through Dunnett test: * *p* < 0.05; ** *p* < 0.01.

**Figure 2 toxins-15-00319-f002:**
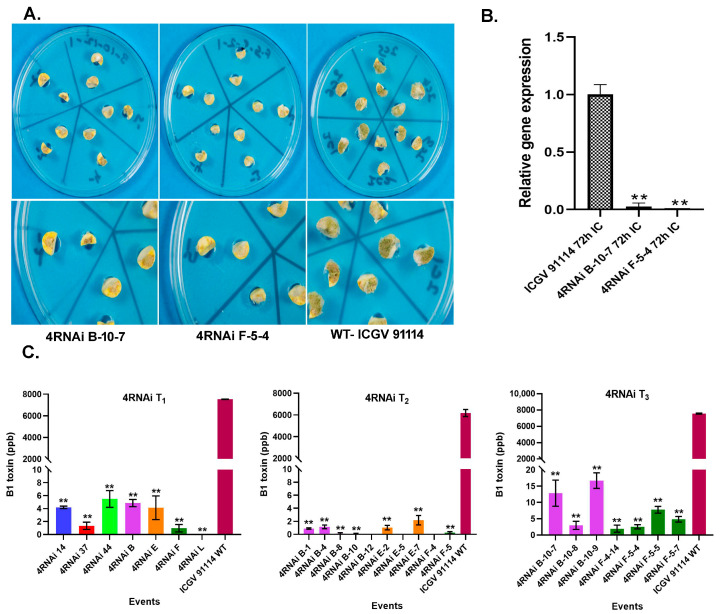
*Aspergillus* growth and sporulation in groundnut control and HIGS 4RNAi lines at 72 h post-infection. (**A**) Screening for fungal colonization on cotyledons of 4RNAi B-10-7, 4RNAi-F-5-4, and WT controls. (**B**) Fungal load of *A. flavus* on cotyledons of T_3_ generation 4RNAi and their WT counterparts. (**C**) Aflatoxin content (ppb) in T_1,_ T_2,_ and T_3_ cotyledons of 4RNAi groundnut lines and untransformed WT controls at 72 hpi. Significant differences between HIGS and wild type control plants were analyzed through Dunnett test: ** *p* < 0.01.

**Figure 3 toxins-15-00319-f003:**
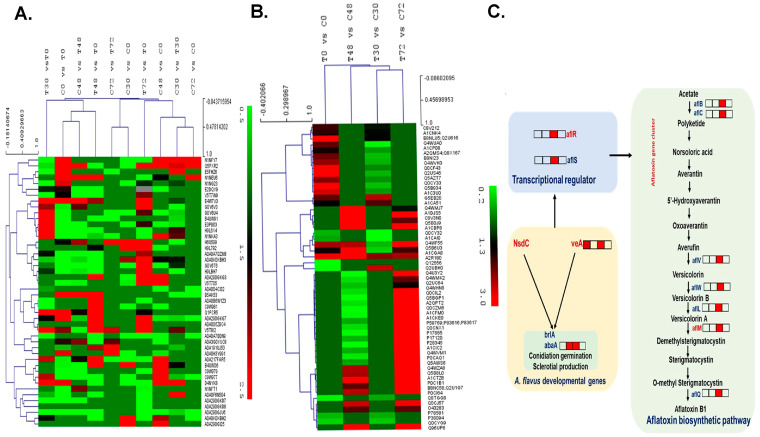
Clustering analysis of significantly differentially expressed proteins (DEPs) in groundnut cotyledons in response to *A. flavus* infection at 0, 30, 48, and 72 hpi. (**A**) Heat map and hierarchical clustering of groundnut proteins differentially expressed in control and HIGS samples at various time points; (**B**) Heat map and hierarchical clustering of *A. flavus* proteins differentially expressed in both control and HIGS samples at various time points; (**C**) Schematic showing the effect of RNAi silencing of *nsdC*, *veA*, *aflR*, and *aflM* genes on developmental genes and aflatoxin biosynthetic pathway regulatory and biosynthetic genes in HIGS and WT-control lines. The downregulated proteins in HIGS lines at 0, 30, 48, and 72 hpi are indicated as red (downregulated) squares.

**Figure 4 toxins-15-00319-f004:**
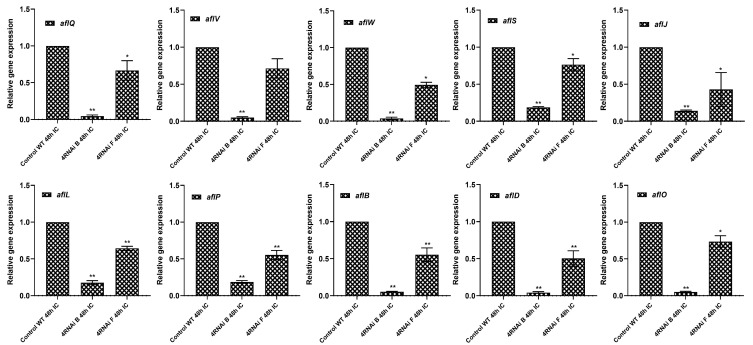
Relative expression of selected *A. flavus* genes that encode differentially expressed proteins in 4RNAi HIGS and WT lines. Quantitative RT-PCR of RNAs isolated from 48 hpi samples were normalized to the *A. flavus* beta-tubulin. Significant differences between HIGS and wild-type control plants were analyzed through Dunnett test: * *p* < 0.05; ** *p* < 0.01.

**Figure 5 toxins-15-00319-f005:**
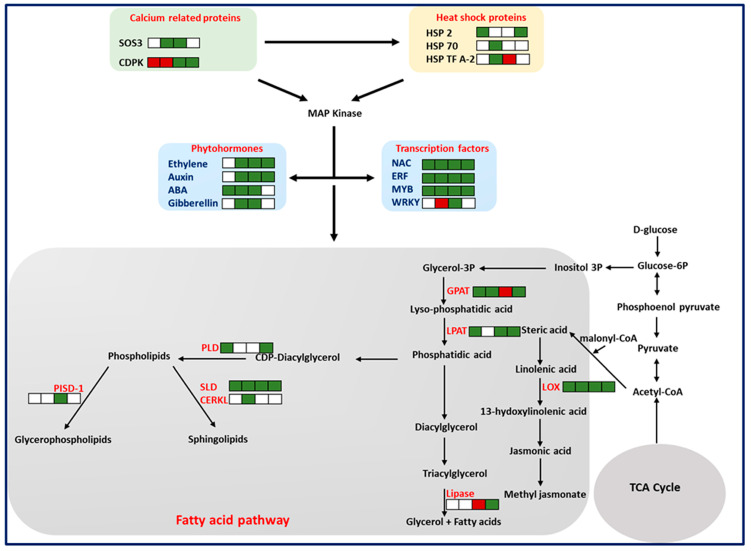
Overview of the *A. flavus*–groundnut interaction showing proteins related to resistance in HIGS lines infected with *A. flavus*. Detailed information on these proteins is shown in Table 2 and Appendix A. Information on proteins related to calcium signaling (Table 2), heat shock proteins (Table 2), phytohormones (Table 2), transcription factors (Table 2), mitogen-activated protein kinase (MAPK), (Appendix A), secondary metabolic pathway genes, lipoxygenase (LOX), glycerol-3-phosphate acyltransferase (GPAT), lysophosphatidyl acyltransferase (LPAT), phospholipase D (PLD), sphingolipid Δ-8 desaturase (SLD), ceramide kinase-related protein (CERKL), and phosphatidylserine decarboxylase proenzyme 1 (PISD-1) are presented in Table 2. While the upregulated and downregulated proteins at 0, 30, 48, and 72 hpi are indicated as green and red squares, the absent proteins are indicated as white squares.

**Figure 6 toxins-15-00319-f006:**
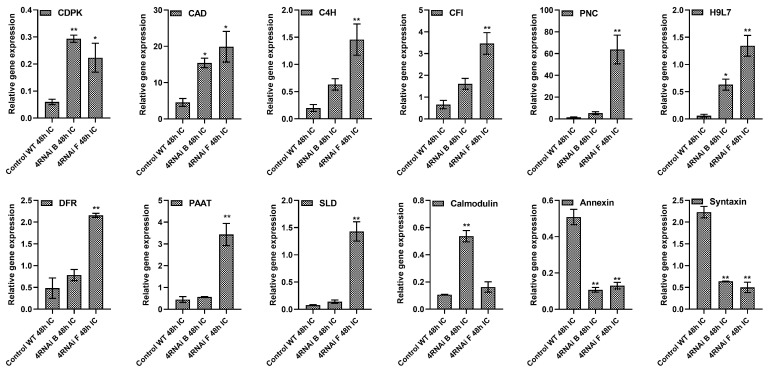
Relative expression of genes encoding differentially expressed proteins from 4RNAi HIGS lines and WT controls. Quantitative RT-PCR of RNAs isolated from 48 hpi samples were normalized to the *A. flavus* ADH3 and G6Pd housekeeping genes: calcium-dependent protein kinase (CDPK), cinnamyl alcohol dehydrogenase (CAD), cinnamic acid 4-hydroxylase (C4H), chalcone-flavanone isomerase (CFI), cationic peroxidase 2 (PNC), CDP-diacylglycerol-glycerol-3-phosphate 3-phosphatidyl transferase (H9L7), dihydroflavonol-4-reductase (DFR), lysophosphatidyl acyltransferase 5 (LPAT), sphingolipid Δ-8 desaturase (SLD), calmodulin, annexin, and syntaxin. Significant differences between resistant and susceptible plants were analyzed through Dunnett test: * *p* < 0.05; ** *p* < 0.01.

**Table 1 toxins-15-00319-t001:** Differential expression of *A. flavus* developmental and aflatoxin biosynthetic pathway-related proteins observed during pathogenesis in 4RNAi-HIGS lines compared to WT controls.

Accession	Mass (kDa)	Protein Name	0 h	30 h	48 h	72 h
A. Fungal differentiation, development, and pathogenicity						
P60204; P60205	17.01	Calmodulin			−46.44	
A0A1R3RGK4	27.39	Ochratoxin biosynthesis cluster transcription factor			19.16	
A2QK82	34.97	Probable pectinesterase A		77.76		
Q2UNJ0	37.39	Chitin synthase export chaperone				−14.57
Q5B8Y3; Q2UQ34	37.54	Eukaryotic translation initiation factor 3 subunit I				−35.84
Q1XGE2; Q8TFU8	37.79	Transcriptional activator hacA				−23.18
Q5ATQ3	39.87	Endopolygalacturonase AN8327				−26.66
Q5AQJ1	39.94	Probable pectin lyase D				21.05
C8VDI2	40.83	Autophagy-related protein 3			1.95	13.92
A1CLZ1	41.63	Diels-Alderase ccsF				13.77
C8VQG9; Q6TLK5	43.31	Secondary metabolism regulator laeA			−10.42	
Q2U6D5	45.27	Autophagy-related protein 18		11.70		
A2QMH1	55.14	Kynurenine 3-monooxygenase 2				−11.03
Q12730	56.63	Protein disulfide-isomerase			−10.25	
Q5B9G5	56.87	Mannitol 2-dehydrogenase				72.84
B0Y7U1; Q4WMR0	58.54	Probable feruloyl esterase B-2			−15.05	
P28298	60.49	Isocitrate lyase	2.21			37.51
A1DG37	66.19	Autophagy-related protein 22-1			−22.57	22.77
A1CEH4	69.49	Vacuolar fusion protein mon1		83.61		
Q12062	70.39	Versicolorin B synthase				−30.71
Q9HFB3; Q96UW0	71.31	pH-response transcription factor pacC/RIM101	10.99		−16.96	
Q2UB56	80.15	Sorting nexin mvp1	−12.61			
B8NBX4	84.06	Cell pattern formation-associated protein stuA				−12.90
P20945	89.34	Conidiophore development regulator abaA		−2.64	−11.04	
Q9R1S8	93.36	Calpain-7	12.10			
Q92197	101.78	Chitin synthase C				20.18
Q4WPF2	106.42	Serine/threonine-protein kinase atg1			−1.57	−12.29
Q00078	123.26	Protein kinase C-like			−4.39	−17.63
B. Aflatoxin biosynthetic pathway						
Q8TGA1	21.22	Fatty acid synthase beta subunit (aflB)		2.15		
P50161	28.15	Versicolorin reductase 1 (aflM)		−4.26		
B9WYE6	38.87	Versiconal hemiacetal acetate reductase (vrdA)			−1.90	
B8NUL8; Q2U4H2	46.30	Lipoyl synthase_ mitochondrial (aflA)		−1.67	−3.75	
O42716	47.63	Aflatoxin cluster transcriptional coactivator (aflS)			−5.71	
P52957	47.25	Sterigmatocystin biosynthesis regulatory protein (aflR)			−2.01	
P0C1B3; P30292	55.32	Alpha-amylase A type-1/2 (amy1)			2.82	
Q6UEF3	55.48	FAD-binding monooxygenase (aflW)			−1.75	−9.20
Q6UEG2	55.65	Cytochrome P450 monooxygenase (aflN)				2.74
Q6UEF1	56.06	Oxidoreductase (AflY)	−2.50			
Q6UEH4	56.23	Cytochrome P450 monooxygenase (aflU)			−8.42	
Q9UW95	56.73	Versicolorin B desaturase (aflL)			−3.79	
Q6UEF4	56.73	Cytochrome P450 monooxygenase (aflV)			−3.92	
Q5BBM1	57.79	Sexual development regulator (velC)			−3.47	
O13345	60.47	O-methyl sterigmatocystin oxidoreductase (aflQ)			−3.46	
E9RCK4	63.20	Developmental and secondary metabolism regulator (veA)	−2.21		−2.44	
Q12062	70.38	Versicolorin B synthase (AflK)				−30.71
Q8TGA1	212.26	Fatty acid synthase beta subunit (aflB)			−1.71	
Q12053	232.94	Norsolorinic acid synthase (aflC)			−7.06	

**Table 2 toxins-15-00319-t002:** Resistance-related proteins identified in the 4RNAi transgenic lines in groundnut *A. flavus* infection.

Accession	Mass	Description	Fold Change against WT Control			
			0 h	30 h	48 h	72 h
Heat shock proteins and calcium signaling-related proteins						
B4UW51	14.52	Class II small heat shock protein Le-HSP17.6			2.50	
B4UW89	18.10	Heat shock protein 2	1.97			2.50
E3NYT2	19.04	Heat shock protein 70		3.62		
A0A068VVA2	24.85	Ca^2+^ hinding-protein SOS3		2.14	2.27	
E7CQA1	40.70	Heat shock transcription factor A-2		2.10		
V5M2Y8	61.53	Calcium-dependent protein kinase			2.28	5.94
Phytohormones						
Q5QET3	8.39	Auxin-induced putative CP12 domain-containing protein		2.40		
Q5QET8	9.69	Auxin-induced putative aldo/keto reductase family protein				4.44
B4UW77	12.57	Gibberellin-regulated protein		1.61	1.94	
E3NYH5	14.39	S-adenosyl methionine synthase			2.00	3.04
M4TG02	30.48	Auxin signaling F-box 3			3.38	
A0A023IUN1	33.82	Abscisic acid 8-hydroxylase 3			8.95	
D7RJM3	39.89	S-adenosylmethionine decarboxylase proenzyme		2.02		
K0FB33	45.18	Ethylene-responsive element binding factor 6			2.03	
G4X5C7	48.26	ABA response element binding protein 1		2.86	12.57	
U6NJF1; K4PM24	55.28	ABA 8′-hydroxylase	4.35	3.19		
Transcription factors						
V5T7X7	8.47	Putative MYB-related protein 25		5.33	1.86	
M4SZY9	22.35	Ethylene-responsive transcription factor	1.64	2.03		66.16
V5T7W6	22.51	Putative R2R3 MYB protein 8		2.10		
A0A0H3Y991	24.02	Wuschel-related homeobox 13B1				3.00
A0A0H3Y7V8	24.25	Wuschel-related homeobox 13A	1.78			
E4W7V3	24.80	Putative DREB transcription factor	1.91		9.65	
M4SZZ4	24.80	Nuclear transcription factor Y subunit A-3		2.03		
M4T2P8	25.06	F-box family protein 6	2.55			
V5T684	25.58	Putative R2R3 MYB protein 9	1.55		2.40	
V5T6N4	27.20	Putative R2R3 MYB protein 1	4.73			
A0A1L1VTR5	32.84	MYB-like transcript factor 6			3.73	
V5T7W9	33.01	Putative MYB-related protein 14			7.39	
V5T688	33.56	Putative MYB-related protein 16		1.58	4.19	
J9Q9Z8	33.74	Ethylene-responsive element binding factor 3			24.17	70.87
V5T6Q5	34.36	Putative MYB-related protein 28		1.91		
K0FBW3	34.41	Ethylene-responsive element binding factor 4	6.26		3.14	
V5T8I2	35.73	Putative R2R3 MYB protein 7		1.85		2.21
V5T714	36.32	Putative MYB-related protein 22			2.66	
C6EU67	37.94	NAC-like transcription factor 3		2.20	4.28	
V5T7Y1	38.12	Putative MYB-related protein 30		1.65	4.50	
C6EU68	39.37	NAC-like transcription factor	1.67		2.62	4.04
B5AK53	39.67	WRKY transcription factor 15			12.15	
V5T8I4	48.21	Putative MYB-related protein 13	1.60		1.99	
Fatty acids						
D3YM77	14.28	Acyl carrier protein		3.92	29.37	
A0A0A6ZDY1	15.56	Peptidyl-prolyl cis-trans isomerase		2.56	2.96	
N1NKF7	15.99	Glycerol-3-phosphate dehydrogenase				22.47
A0A0A6ZDP1	20.10	Glyceraldehyde-3-phosphate dehydrogenase C2		2.04	3.26	
B4UW57	24.67	Putative dihydroflavonol reductase	1.67			
B4UW49	26.61	Putative lipase				1.58
B4UWB9	26.79	Lipoxygenase 1			7.97	
A0A0U3E0B1	27.41	Phosphatidyl inositol phosphate kinase		1.98	4.68	
D8KXY5	41.11	Malonyl-CoA:ACP transacylase 1-1			3.86	
A0A384QWC2	43.91	Acyl-[acyl-carrier-protein] desaturase		2.88	1.86	6.09
A0A3G0YUC8	44.36	Lysophosphatidyl acyltransferase 5	5.40		3.90	1.82
A0A384QZQ3	45.64	Palmitoyl-monogalactosyldiacylglycerol delta-7 desaturase		1.97	6.01	
N1NG06	48.15	3-ketoacyl-CoA synthase		1.67	3.48	
N1NFY7	48.58	Putative ceramide kinase-related protein		3.18		
A0A0R4UXQ1	48.86	3-ketoacyl-CoA thiolase			4.58	
A0A0R4VXV1	50.46	Phosphatidylserine decarboxylase proenzyme 1			3.59	
E6Y9A7	50.47	Beta-ketoacyl-ACP synthetase I	1.67			5.86
A0A384QZP9	52.49	Sphingolipid delta8 desaturase	2.29	2.22	2.39	9.71
A0A0K0K9Q6	53.75	Glyceraldehyde-3-phosphate dehydrogenase	1.82		3.87	3.85
A0A0R4UXP7	56.05	Glycerol-3-phosphate acyltransferase 6	3.25	2.53		
A0A385I5T0	57.54	3-ketoacyl-CoA synthase	2.36			
A0A109Z9U2	75.55	Long chain acyl-CoA synthetase 1	1.51		2.02	
F1AM70	79.43	Triacylglycerol lipase 1				1.65
A0A0R4VUF1	91.12	Digalactosyldiacylglycerol synthase 1	2.61			
Q2HWT7	91.27	Phospholipase D	2.63			4.15
Q4JME6	97.76	Lipoxygenase	1.63	9.88	3.43	3.36

**Table 3 toxins-15-00319-t003:** Susceptibility-associated proteins identified in transgenic 4RNAi groundnut lines infected with *Aspergillus flavus*.

Accession	Mass	Description	Change against WT at Different Times after Infection (In Folds)			
			0 h	30 h	48 h	72 h
E2DQY9	8.92	Heat shock protein DnaJ			−2.57	−3.69
B4UW90	12.43	Heat shock protein 3			−1.58	
B1PMD1	13.50	Zinc finger protein ZFP133		−1.87		
Q6R2U6	16.64	Calmodulin		−3.33		
Q06H39	17.10	Syntaxin	−2.05	−1.93		−4.02
B4UWB2	22.09	Kunitz trypsin inhibitor 4		−1.64		
B4UW91	25.92	Putative heat shock protein 4		−1.53	−11.67	−3.21
V5T7W3	27.81	Putative R2R3 MYB protein 3		−1.63	−2.60	
E3NYG8	28.02	Zinc finger protein, ZAT10-like			−3.51	
V5T6P8	28.83	Putative MYB-related protein 18	−1.85	−2.49	−1.71	
B5AK52	30.03	NAC-like transcription factor 2			−5.56	−4.30
V5T692	32.43	Putative MYB-related protein 21		−1.51	−5.59	
V5T8J4	32.74	Putative MYB-related protein 29			−1.83	−2.05
B2ZHY3	34.34	NAC-like transcription factor		−1.83	−1.68	
A0A0F6VX63	36.25	Annexin			−6.26	
V5T7X2	37.51	Putative MYB-related protein 20		−3.00	−2.97	
K0FAV2	41.50	Ethylene-responsive element binding factor 1		−2.31	−2.70	−2.52
D8KXZ7	41.53	Enoyl-ACP reductase 1–2		−2.00	−6.41	
E6Y9A8	52.10	Chloroplast omega-6 fatty acid desaturase	−2.96	−1.63	−1.95	
A0A290G010	56.63	MLO-like protein		−1.60	−3.82	
A0A385I5T0	57.54	3-ketoacyl-CoA synthase			−2.16	
B4YA12	58.21	Calcium calmodulin-dependent protein kinase			−2.28	
D8KXZ0	59.54	Beta-ketoacyl-ACP synthase II-1			−41.34	
Q70KY0	67.36	9-cis-epoxy carotenoid dioxygenase			−1.89	−7.53
A0A109QJM5	75.06	Long chain acyl-CoA synthetase 4			−12.32	
N1NFY2	105.41	Putative Zinc finger_ C3HC4 type (RING finger)			−7.70	

## Data Availability

All data are presented in figures. The mass spectrometry proteomics data are deposited at the ProteomeXchange consortium via jPOSTrepo with the database identifier PXD028196.

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
