# Peer review of "Multiplexed Host-Induced Gene Silencing of Aspergillus flavus Genes Confers Aflatoxin Resistance in Groundnut"

_toxins, 2023, doi:10.3390/toxins15050319_

Round 1

Reviewer 1 Report

In this work, multiplexed host-induced gene silencing (HIGS) of Aspergillus flavus genes essential for fungal sporulation and aflatoxin production confer enhanced resistance to Aspergillus infection and aflatoxin contamination in groundnut. This is a good strategy for peanuts to resist fungal contamination. But the experimental design is limited and the following issues need to be considered.

1. Most of the result Figures are not clear, for example, figure 2, it is difficult to observe differences between the differential wild type and 4RNAi B-10-7, 4RNAi F-5-4, WT-ICGV91114 strains after Aspergillus infestation.

2. No results to support no any noticeable effects in HIGS plants.

3. In T1, T2, and T3 cotyledons of 4RNAi groundnut lines, aflatoxin tested at no more than 20 ppb at 72 hours, what about seven days. Do the results still support the conclusions?  Please add results or qualify the conclusion.

Author Response

Reviewer 1

  1. Most of the result Figures are not clear, for example, figure 2, it is difficult to observe differences between the differential wild type and 4RNAi B-10-7, 4RNAi F-5-4, WT-ICGV91114 strains after Aspergillus

Response: We have updated Figure 2A to show the inhibition of fungal growth in HIGS lines compared to wild type control. Along with this we have also updated other figures in the manuscript for more clarity.

  1. No results to support any noticeable effects in HIGS plants.

Response: We observed HIGS lines have significantly lower fungal biomass and aflatoxin production compared to wild type genotypes. Further, we did not observe any noticeable morphological changes in HIGS lines compared to its wild types. This has been shown in Supplementary Figure S1.

  1. In T1, T2, and T3 cotyledons of 4RNAi groundnut lines, aflatoxin tested at no more than 20 ppb at 72 hours, what about seven days. Do the results still support the conclusions?  Please add results or qualify the conclusion.

Response: Aflatoxin was quantified in HIGS and wild type genotypes at 72 hpi as described by Arias et al., 2015. We observed consistent results at 72 hpi, as seeds get degraded by 96 hpi.

-----------

Reviewer 2 Report

The authors produced and analyzed the mutant strains of groundnut expressing RNAi cassettes of nsdC, veA, aflM and aflR from Aspergillus flavus. The results in this manuscript clearly indicated that these groundnut mutants strongly suppressed growth and aflatoxin production of A. flavus. Furthermore, proteome analyses of the groundnuts mutants and the fungal strain were also carried out. I think this study attracts the reader’s interest. However, this manuscript has several issues.

[Major points]

Line 14 and Line 179

It is not “AbaB” but “AbaA” has been reported as a conidiation regulator of A. flavus (Cho, H. et al., Regulation of Conidiogenesis in Aspergillus flavus. Cells 2022, 11, 2796). Although the recode of “Q2U9L6” indicates that AbaB is a conidiation regulator in A. oryzae RIB40, function of AbaB orthologs have not been characterized in Aspergillus species (including A. flavus and A. oryzae) experimentally. AbaB may not be concerned with regulation of conidiation. Therefore, I strongly recommend that the authors delete the descriptions of “AbaB”.

Figure 1

This figure is too small to see (Especially 1A and 1B). This prevents the understanding of the RNAi cassettes used in this study. I recommend to enlarge this figure.

Supplemental Figure S1

This figure is also too small to read the characters in this figure.

Statistical analysis of the data in Figure 4 and Figure 6.

Student’s t-test should not be used in multiple sample data.

In these cases, I recommend to use “dunnett test”.

Statistical analysis of the data in Figure 1E

Recently, Duncun’s test is not preferred. Instead, dunnett test and tukey's test are recommend to use.

Figure 2A

The authors reported that growth and conidiation inhibition of A. flavus strain when the fungus was grown on the HIGS groundnuts. But these phenomena are head to recognize from Figure 2A. If possible, in addition to the photograph of Figure 2A, I recommend that the authors show the photographs of microscopic observations (using a stereo microscope and a digital microscope etc.) of the samples.

[Minor points]

Line 48

48 hpi samples -> 48 hpi (hours post infection) samples

Line 305

nsdC -> nsdC

Line 416

SiCML55 in tomatoes inhibits Phytophithora infraction [59].

-> SiCML55 in tomatoes inhibits Phytophithora infraction [59].

[Other comments]

Can the HIGS groundnuts produced in this study inhibit the growth and aflatoxin production of Aspergillus parasiticus?

Are the defense systems against fungi activated in the HIGS groundnuts before infection of A. flavus? (Does the expression of the RNAi cassettes affect the defense system in the RNAi groundnuts?)

Author Response

Reviewer 2

  1. Line 14 and Line 179: It is not “AbaB” but “AbaA” has been reported as a conidiation regulator of  flavus(Cho, H. et al., Regulation of Conidiogenesis in Aspergillus flavusCells 2022, 11, 2796). Although the recode of “Q2U9L6” indicates that AbaB is a conidiation regulator in A. oryzae RIB40, function of AbaB orthologs have not been characterized in Aspergillus species (including A. flavus and A. oryzae) experimentally. AbaB may not be concerned with regulation of conidiation. Therefore, I strongly recommend that the authors delete the descriptions of “AbaB”.

Response: Data on AbaB has been deleted based on reviewers’ comments.

  1. Figure 1: This figure is too small to see (Especially 1A and 1B). This prevents the understanding of the RNAi cassettes used in this study. I recommend to enlarge this figure.

Response: Figure 1 has been modified based on reviewer suggestions.

  1. Supplemental Figure S1: The figure is too small to read the characters in this figure.

Response: Supplementary Figure S1 (Now Supplementary Figure S4) has been modified based on reviewer suggestions.

  1. Statistical analysis of the data in Figure 4 and Figure 6: Students t-test should not be used in multiple sample data. In these cases, I recommend using “Dunnett test”.

Response: As reviewer suggested, we have updated Figure 4 and Figure 6 using Dunnett test.

  1. Statistical analysis of the data in Figure 1E: Recently, Duncun’s test is not preferred. Instead Dunnett test and tukey’s test are recommend to use.

Response: As reviewer suggested, we have updated Figure 1E using Dunnett test.

  1. Figure 2A: The authors reported that growth and conidiation inhibition of  flavusstrain when the fungus was grown on the HIGS groundnuts. But these phenomena are hard to recognize from Figure 2A. If possible, in addition to the photograph of Figure 2A, I recommend that the authors show the photographs of microscopic observations (using a stereo microscope and a digital microscope etc.) of the samples.

Response: Microscopic images showing the fungal growth and conidia inhibition between HIGS and wild types are included in Supplementary Figure S2.  

  1. Minor points
    1. Line 48: 48 hpi samples -> 48 hpi (hours post infection) samples
    2. Line 305: nsdC-> nsdC
    3. Line 416: SiCML55 in tomatoes inhibits Phytophithora infraction [59]. -> SiCML55 in tomatoes inhibitsPhytophithora infraction [59].

Response: We have modified the text as per the reviewer’s suggestion.

  1. Other comments: Can the HIGS groundnuts produced in this study inhibit the growth and aflatoxin production of Aspergillus parasiticus?

Response: We tested HIGS groundnut lines with only A. flavus strain Af11-4. We observed the inhibition of fungal growth and aflatoxin production. But we did not test the HIGS lines for Aspergillus parasiticus.

  1. Are the defense systems against fungi activated in the HIGS groundnuts before infection of  flavus? (Does the expression of the RNAi cassettes affect the defense system in the RNAi groundnuts?)

Response: Yes, we observed the defense systems activated in HIGS groundnut compared with its wild type before infection of A. flavus (i.e. 0 hpi). We speculated that this constitutive defense response is due to expression of the RNAi cassettes in the groundnut genome.  

------------

Reviewer 3 Report

In this manuscript, the authors showed HIGS effect on target genes and inhibitory effect on A. flavus growth and aflatoxin production on cotyledons produced in this study. The results are clear. And the authors were examined the expression profile during the interaction between host and A. flavus using DEP analysis and qRT-PCR. These are well-described. I think the study is interested in researchers in this field. I have some minor concerns and described them.

1.  In my view, the resolution of the figures should be increased and the letters should be enlarged. And the upper left caption "Figure x" could be removed.

2. Such as Line 18 and Line 32, excess spaces were found in the manuscript. The authors need to check the main text again.

3. Regarding the sentence "Fungal differentiation and pathogenicity proteins including calmodulin, transcriptional activator-hacA, kynurenine 3-monooxygenase 2, veA, velC, abaB, and several aflatoxin pathway biosynthetic enzymes ..." in Line 13 to 15. The authors stated that they were protein. If so, for example, "hacA" should be "HacA". In the same way, VeA, velC, abaB, ..., they are better to described as proteins. Many symbols of genes and proteins were used in the manuscript. In the revision, the authors need to check them again. 

Author Response

Reviewer 3

  1. In my view, the resolution of the figures should be increased, and the letters should be enlarged. And the upper left caption "Figure x" could be removed.

Response: We have updated the figures as per reviewers’ suggestions.

  1. Such as Line 18 and Line 32, excess spaces were found in the manuscript. The authors need to check the main text again.

Response: We thank reviewer for pointing this out. The excess spaces were removed in the text.

  1. Regarding the sentence "Fungal differentiation and pathogenicity proteins including calmodulin, transcriptional activator-hacA, kynurenine 3-monooxygenase 2, veA, velC, abaB, and several aflatoxin pathway biosynthetic enzymes ..." in Line 13 to 15. The authors stated that they were protein. If so, for example, "hacA" should be "HacA". In the same way, VeA, velC, abaB, ..., they are better to described as proteins. Many symbols of genes and proteins were used in the manuscript. In the revision, the authors need to check them again.

Response: As reviewer suggested, the following corrections have been made: hacA to HacA;       veA to VeA;  velC to VelC; abaB to AbaB.

-----------

Round 2

Reviewer 1 Report

The quality of this manuscript has been improved after revision. I think it is accepted in the present form.

Reviewer 2 Report

The authors investigated the RNAi groundnuts strains that was able to inhibit the growth and aflatoxin production of Aspergillus flavus. Furthermore, proteome analyses of the groundnut mutants and A. flavus was also carried out. 

They were several issues in this manuscript in the previous version. In the revised version, the authors properly addressed these issues.